# Identification and Total Synthesis of Two Previously Unreported Odd-Chain Bis-Methylene-Interrupted Fatty Acids with a Terminal Olefin that Activate Protein Phosphatase, Mg^2+^/Mn^2+^-Dependent 1A (PPM1A) in Ovaries of the Limpet *Cellana toreuma*

**DOI:** 10.3390/md17070410

**Published:** 2019-07-11

**Authors:** Hideki Kawashima, Naoki Toyooka, Takuya Okada, Huy Du Nguyen, Yuya Nishikawa, Yuka Miura, Nana Inoue, Ken-ichi Kimura

**Affiliations:** 1Bioscience Laboratory, Miyako College Division, Iwate Prefectural University, Miyako, Iwate 027-0039, Japan; 2Graduate School of Innovative Life Science, University of Toyama, 3190 Gofuku, Toyama 930-8555, Japan; 3Graduate School of Science and Engineering University of Toyama, 3190 Gofuku, Toyama 930-8555, Japan; 4Chemical Biology Laboratory, Graduate School of Arts and Sciences, Iwate University, Morioka, Iwate 020-8550, Japan

**Keywords:** non-methylene-interrupted fatty acids, bis-methylene-interrupted dienoic acids, limpet, *Cellana toreuma*, cytotoxicity, (12*Z*)-12,16-heptadecadienoic acid, (14*Z*)-14,18-nonadecadienoic acid, equivalent chain length, protein phosphatase, Mg^2+^/Mn^2+^-dependent 1A (PPM1A)

## Abstract

Diverse non-methylene-interrupted (NMI) fatty acids (FAs) with odd-chain lengths have been recognized in triacylglycerols and polar lipids from the ovaries of the limpet *Cellana toreuma*, however their biological properties remain unclear. In this study, two previously unreported odd-chain NMI FAs, (12*Z*)-12,16-heptadecadienoic (**1**) and (14*Z*)-14,18-nonadecadienoic (**2**) acids, from the ovary lipids of *C. toreuma* were identified by a combination of equivalent chain length (ECL) values of their methyl esters and capillary gas chromatography-mass spectrometry (GC-MS) of their 3-pyridylcarbinol derivatives. On the basis of the experimental results, both **1** and **2** were synthesized to prove their structural assignments and to test their biological activity. The ECL values and electron impact-mass (EI-MS) spectra of naturally occurring **1** and **2** were in agreement with those of the synthesized **1** and **2**. In an in vitro assay, both **1** and **2** activated protein phosphatase, Mg^2+^/Mn^2+^-dependent 1A (PPM1A) up to 100 μM in a dose-dependent manner.

## 1. Introduction

Non-methylene-interrupted (NMI) fatty acids (FAs) with two or more methylene groups between the double bonds are frequently distributed in marine invertebrates [1,2], including mollusks and sponges, as well as in some terrestrial plant seeds [3,4], however their biological role and function are poorly understood. Among representative marine NMI FAs in sponges, some typical Δ5,9-dienoic FAs have inhibitory activities against key enzymes targeted for DNA uncoiling and cleavage and FA synthesis of some bacterial cells [1,2], as well as cytotoxicity against some bacteria and some cancer cell lines [1,2]. Among several Δ5,9-dienoic FAs and their related derivatives which are targeted enzyme inhibitors, (5*Z*,9*Z*)-5,9-heptacosadienoic acid from the marine sponge *Amphimedon* sp. shows the most potent inhibitory activity against topoisomerase [1,2]. In addition, a mixture of (5*Z*,9*Z*)-5,9-tricosadienoic and (5*Z*,9*Z*)-5,9-tetracosadienoic acids from the marine sponge *Agelas oroides* inhibited enoyl-acyl carrier protein reductase in fatty acid biosynthesis [2]. However, other biological properties of NMI dienoic FAs are still unknown and their amounts obtained are generally insufficient for an in vitro targeting bioassay.

Although little is known about the occurrence and distribution of uncommon NMI FAs in marine mollusks, the highly structural diversity of NMI FAs, including their positional isomers, is found in ovaries of dominant species of limpets *Cellana grata*, *Cellana toreuma,* and *Collisella dorsuosa* belonging to marine archaeogastropods [5,6,7,8]. As an unexpected finding, uncommon NMI dienoic FAs in the ovaries of *C. toreuma* especially comprised of diverse Δ5,9-dienoic and Δ9,15-dienoic FAs, along with several minor previously undescribed FAs. Interestingly, these Δ5,9-dienoic FAs, some of which have been reported for their biological activity in sponges, are also recognized in limpet gonads [9]. Despite the structural diversity of NMI FAs, their biological properties have not yet been investigated. Among our identified NMI FAs in limpets, (9*Z*,20*Z*)-9,20-tricosadienoic acid (23:2Δ9*Z*,20*Z*) shows an in vitro glycogen synthase kinase (GSK)-3β inhibitory activity and cytotoxicity against HL60 cells, with IC_50_ values of 8.7 and 125.8 μM, respectively [10]. This finding could be a useful clue to define the biological activity of marine uncommon NMI dienoic FAs. Interestingly, as a structural analogue of 23:2Δ9,20, (4*Z*,15*Z*)-4,15-octadecadienoic acid (18:2Δ4*Z*,15*Z*), with inhibitory activity against GSK-3β with an IC_50_ value of 21.9 μM, shows an antidiabetic effect on rat hepatoma H4IIE cells [10]. This compound also displayed a neurite outgrowth activity on Aβ(25–35)-treated rat cortical neurons [11]. From this finding, the unusual structure of 18:2Δ4*Z*,15*Z* with dual biological functions could be useful to design biologically active compounds, such as small functional molecules, to increase or improve targeting health effects in the future. Recently, in our study on the identification of previously undescribed dienoic FAs (with less than 0.1% of the total FAs) in ovaries of *C*. *toreuma*, we have identified six NMI dienoic FAs with a terminal olefin, namely, 5,16-heptadecadienoic (17:2Δ5,16), 7,16-heptadecadienoic (17:2Δ7,16), 7,18-nonadecadienoic (19:2Δ7,18), 11,18-nonadecadienoic (19:2Δ11,18), 7,20-heneicosadienoic (21:2Δ7,20), and 11,20-heneicosadienoic (21:2Δ11,20) acids, along with three putative precursors, 14-pentadecenoic (15:1Δ14), 16-heptadecenoic (17:1Δ16) and 18-nonadecenoic (19:1Δ18) acids, and one known analogue 9,18-nonadecadienoic acid (19:2Δ9,18) [12,13]. Among these NMI FAs, the structural assignments of minor naturally occurring 19:2Δ7*Z*,18 and 21:2Δ7*Z*,20 were finally confirmed by chemical synthesis through the Wittig reaction under salt-free conditions for selective formation of internal *Z*-double bonds [14].

Protein phosphatase, Mg^2+^/Mn^2+^-dependent 1A (PPM1A, also named PP2Cα) regulates several signaling pathways including the p38, JNK, Wnt, and p53 networks [15]. In addition, as a key negative regulator, PPM1A dephosphorylates Smad2/Smad3 to block the TGF-β signaling pathway [16]. Since PPM1A deficiency is involved in several diseases, PPM1A activation by small molecules might be a promising strategy for the treatment of cancer and/or liver fibrosis [17,18]. As one of the potent PPM1A activators [19], 18:1n-9 causes the induction of apoptosis in neuronal and endothelial cells [20,21].

The occurrence of NMI dienoic FAs with a terminal olefin can contribute to the structural diversity of marine NMI FAs, as well contribute to a better understanding of their biological functions. This study reports the identification and total synthesis of previously unreported NMI dienoic FAs **1** and **2** that activate PPM1A.

## 2. Results and Discussion

### 2.1. Identification of ***1*** and ***2*** in Ovaries of the Limpet Cellana toreuma

In this study, a total of 65 different FAs, along with diverse NMI di-, tri-, and tetraenoic FAs, which we have identified previously, were detected in ovaries of *C. toreuma*, except for some minor FAs (less than 0.1% of the total FAs). Among the identified FAs, 16:0, 18:1n-9, 18:1n-11, 20:4n-6, and 20:5n-3 were major components, whereas 20:4n-6 and 20:5n-3 (more than 15% of the total FAs) were present in polar lipids [9]. Among the previously undescribed minor FAs, **1** and **2** (Figure 1) from triacylglycerols (TAG) and polar lipids were fractionated by using 5% (*w*/*v*) argentation thin-layer chromatography (TLC), however, they were not isolated due to their minute quantity. Despite their minute amounts, **1** and **2** in TAG and polar lipids were detected by gas chromatography-mass spectrometry (GC-MS).

In Figure 2, both **1** and **2**, as minor and uncommon NMI FA components, were detected along with previously described odd-chain isomers, three NMI heptadecadienoic, and two NMI nonadecadienoic acids [12,13], although, 9,16-heptadecadienoic acid (17:2Δ9,16), which is a structural analogue of 19:2Δ11,18, has been recognized in biological samples for the first time in this study. Generally, most NMI FAs are found as minor lipid components as compared with methylene-interrupted FAs, i.e., with the presence of one methylene group between the double bonds.

The enriched fraction including the target methyl esters of **1** and **2** (Figure 2) was submitted to GC-MS analysis and chemical derivatization reaction with 3-pyridylcarbinol.

The EI-MS spectra of both methyl esters yielded a series of diagnostic fragment ions at *m*/*z* 74, [M − 74]^+^ (due to the loss of the McLafferty ions), [M − 32]^+^ and [M]^+^ (Figure 3A and Figure 4A). The EI-MS spectra of the methyl esters of naturally occurring and synthesized **1** showed a molecular ion at *m*/*z* 280 and characteristic fragment ions at *m*/*z* 74, 206 [M − 74]^+^ and 249 [M − 32]^+^, corresponding to the molecular weight of a heptadecadienoic acid. Similarly, in the EI-MS spectra of the methyl esters of naturally occurring and synthesized **2** (Figure 3B and Figure 4B), a series of diagnostic fragment ions at *m*/*z* 74, 234 [M − 74]^+^, and 277 [M − 32]^+^ for FA methyl ester, along with a molecular ion at *m*/*z* 308, were observed, indicating the molecular weight of a nonadecadienoic acid. Equivalent chain length (ECL) values for the methyl esters of naturally occurring **1** and **2** were 17.93 and 19.93, respectively. These results suggested that both **1** and **2** were odd-chain analogues. In this context, the ECL values of the methyl esters of **1** and **2**, obtained by synthesis, were identical to those of naturally occurring **1** and **2**.

To determine the double bond positions of **1** and **2**, their methyl esters were directly converted into their 3-pyridylcarbinol derivatives. The EI-MS spectrum of **1** showed a series of diagnostic fragment ions at *m*/*z* 316 [M − 41]^+^, 328 [M − 29]^+^, 342 [M − 15]^+^ and 356 [M − 1]^+^, together with a molecular ion at *m*/*z* 357, and a gap of 26 amu between *m*/*z* 276 and *m*/*z* 302, indicating the double bond is located on C-12 (Figure 5A). As described previously in the EI-MS spectra of 3-pyridylcarbinol derivatives of NMI FAs with a terminal olefin [12], the presence of key fragment ions at *m*/*z* 316 [M − 41]^+^ and *m*/*z* 328 [M − 29]^+^ for NMI FAs with a terminal olefin was suggested in this study. In addition, the presence of the second abundant fragment ion at *m*/*z* 316 [M − 41]^+^, along with a relatively dominant ion at *m*/*z* 356 [M − 1]^+^ rather than *m*/*z* 357 [M]^+^, as well as a gap of 26 amu between *m*/*z* 276 and *m*/*z* 302, suggested that there was a bis-methylene-interrupted dienoic FA with a terminal olefin. To further confirm the presence of these key diagnostic fragment ions as described above, the EI-MS spectrum of the 3-pyridylcarbinol derivative of **1** was analyzed. Consequently, all the diagnostic ions of **1**, obtained by synthesis (Figure 5B), agreed well with those of naturally occurring **1**. On the basis of these characteristic fragment ions, naturally occurring **1** was finally confirmed as 12, 16-heptadecadienoic acid, namely, bis-methylene-interrupted heptadecadienoic acid with a terminal olefin.

Similarly, in the EI-MS spectrum of the 3-pyridylcarbinol derivative of 2 for both naturally occurring and obtained by synthesis (Figure 6), a series of diagnostic ions for FAs with a terminal olefin were also observed at *m*/*z* 344 [M − 41]^+^, *m*/*z* 356 [M − 29]^+^, *m*/*z* 370 [M − 15]^+^ and *m*/*z* 384 [M − 1]^+^, while a molecular ion at *m*/*z* 385, corresponding to the molecular ion of a nonadecadienoic acid, was recognized together with a gap of 26 amu between *m*/*z* 304 and *m*/*z* 330, indicating the presence of a double bond at C-14. From the above mentioned EI-MS spectral data, naturally occurring **2** was identified as 14,18-nonadecadienic acid, another previously unreported bis-methylene-interrupted dienoic acid with a terminal olefin.

ECL values are practically used to identify a dienoic FA (as its methyl ester), because the ECL of a dienoic FA depends, in principle, on the contribution of the two constitutive ethylenic bonds between the double bonds [22]. In this study, ECL values for the methyl esters of naturally occurring **1** and **2** were 17.93 (calculated ECL 17.92 = ECL minus base value 0.39 for 17:1Δ12*Z* + ECL minus base value 0.53 for 17:1Δ16 + 17.00) and 19.93 (calculated ECL 19.92 = ECL minus base value 0.38 for 19:1Δ14*Z* + ECL minus base value 0.54 for 19:1Δ18 + 19.00), respectively (Table 1). Judging from their ECL values, the double bond of both bis-methylene-interrupted structures was tentatively proposed to have a *Z* configuration. (Therefore, to further determine biological properties of naturally occurring **1** and **2**, as well as to elucidate their precise structures, we have synthesized these compounds.)

In this study, the ECL values for methyl esters and 3-pyridylcarbinol derivatives of **1** and **2**, obtained by synthesis, agreed well with those of naturally occurring **1** and **2**, indicating that the double bond in these compounds has a *Z* configuration. Thus, the structures of **1** and **2** were finally established as (12*Z*)-12,16-heptadecadienoic and (14*Z*)-14,18-nonadecadienoic acids, respectively. With the EI-MS spectra for the 3-pyridylcabiol derivatives of bis-methylene-interrupted dienoic FAs with a terminal olefin, predominant characteristic ions for *m*/*z* [M − 41]^+^ and *m*/*z* [M − 1]^+^ of both **1** and **2**, obtained by synthesis, were unequivocally confirmed in this study. This new knowledge will be very helpful in the future to clarify the structural assignment for undescribed bis-methylene-interrupted FA analogues having a terminal olefin.

### 2.2. Total Synthesis of ***1*** and ***2***

The total synthesis of **1** and **2** started with commercially available diols **3a** and **3b**, respectively. Monoprotection of **3a** and **3b,** with chloromethyl methyl ether (MOMCl), provided the mono-MOM ethers **4a** and **4b**, which were converted to the aldehydes **5a** and **5b** by pyridinium chlorochromate (PCC) oxidation. The Wittig reaction of **5a** and **5b** with the Wittig reagent under salt-free conditions [23] provided the dienes **6a** and **6b** in 84% and 94% yield, respectively. Deprotection of the MOM protecting group in **6a** and **6b**, followed the Jones oxidation of the resulting alcohols, gave rise to **1** and **2** (Scheme 1).

### 2.3. Biological Activity of ***1*** and ***2***

Figure 7 shows the result of PPM1A activation activity of **1** and **2**, obtained by synthesis, and structurally related FAs. All four uncommon NMI FAs, **1**, **2**, 18:2Δ4*Z*,15*Z*, and 23:2Δ9*Z*,20*Z*, displayed PPM1A activation activity up to 100 μM in a dose-dependent manner. However, only 10-undecenoic acid had no activation activity. Compounds **1** and **2** showed a maximum PPM1A activation activity of 175 ± 7.4% and 169 ± 7.1%, at 100 μM, respectively, as compared with 18:1n-9 and 18:2n-6 which are known PPM1A activators [19]. In contrast, **1**, **2,** and 18:1n-9 are inactive against HL60 cells even at a concentration of 130 μM for 48 h incubation (IC_50_ > 130 μM), although in the previous study 18:2Δ4*Z*,15*Z*, 23:2Δ9*Z*,20*Z*, and 18:2n-6 showed cytotoxicity with IC_50_ values of 115.1, 125.8, and 23.0 μM, respectively [10]. In this study, the activation of PPM1A by NMI FAs was demonstrated for the first time, since, so far, 18:1n-9 was found to be the most active activator which causes the induction of apoptosis in neuronal and endothelial cells [20,21]. Clarification of the PPM1A activation based on characteristic structural features of marine uncommon NMI FAs is needed to further investigate their biological properties.

Unfortunately, little is known about the structural diversity and biological properties of NMI FAs in marine mollusks. Our continuing work with highly diverse uncommon NMI FAs, especially their odd-chain analogues and isomers in minute amounts in germ cells of Japanese limpets, has resulted in the discovery of structurally interesting and biologically active **1** and **2**, along with 23:2Δ9*Z*,20*Z*, as novel PPM1A activators, although their functions are still to be elucidated. We expect that studies on the isolation and identification of marine undescribed NMI FAs will provide a good starting point for further investigations of their biological functions, as well as to explore small molecules to increase or improve targeting health effects. Furthermore, a combination of chemically synthesis and structural studies on previously undescribed NMI FAs could yield important findings for their unreported biological properties.

## 3. Experimental Section

### 3.1. General Experimental Procedures

General experimental procedures for all chemical transformations and methods for the determination of the structure of synthesized compounds were performed as described in our former paper including the apparatus for the measurement of ^1^H and ^13^C NMR, IR, and low- and high-resolution electron impact (EI) mass spectra [24]. The copies of ^1^H and ^13^C NMR spectra for all new synthetic compounds (**4b**, **6a**, **6b**, **1**, and **2**) are included in the attached Appendix A.

### 3.2. Collection of Biological Materials, Extraction, Fractionation, and Structural Determination of NMI FAs

Matured *Cellana toreuma* (Nacellidae) was collected on 5 and 25 August 2016 by hand in Otsuchi Bay (latitude 39°20.34′364″N, longitude 141°54.26′931″E), Iwate, northeastern Japan. The samples (40.3–52.5 mm, shell length) were identified by biologist Taiji Kurozumi, Natural History Museum and Institute, Chiba, Japan. A voucher specimen CBM-ZM 179324 was stored at the Institute.

Fresh ovaries of *C. toreuma* were dissected. Polled ovaries (15 g wet weight) were suspended in 50 mL of CHCl_3_-MeOH (2:1, *v*/*v*) and were homogenized for 1 min at 16,000 rpm by using IKA Ultra-Turrax T25 basic (IKA Japan, Nara, Japan). Lipids were extracted from homogenized ovaries by the Bligh and Dyer method [25]. The fractionation and identification of TAG and polar lipids by TLC were performed according to our previous experimental procedure [8]. The amounts of TAG and polar lipids (mainly phosphatidylcholine and phosphatidylethanolamine) obtained were 104 mg and 56 mg, respectively.

FA methyl esters were prepared as previously described [13]. FA methyl esters, depended on the degree of FA unsaturation, were obtained by using 5% (*w*/*v*) argentation TLC, as described previously [13]. The fractionated FA methyl esters, including methyl esters of **1** and **2**, in TAG and polar lipids were approximately 3 mg and 1 mg, respectively. These FA methyl esters were subjected to gas-liquid chromatography (GLC) and GC-MS analyses and chemical derivatization reaction with 3-pyridylcarbinol [13].

GC-MS analyses of FA methyl esters and 3-pyridylcarbinol derivatives, as well as GLC analyses of FA methyl esters, were performed according to our previous works [12,13].

### 3.3. Identification of ***1*** and ***2*** by GC-MS of their Methyl Esters and 3-Pyridylcarbinol Derivatives

Naturally occurring **1**: EI-MS spectrum for methyl 12,16-heptadecadienoate: 280 ([M]^+^, 3), 248 ([M − 32]^+^, 7), 220 (1), 206 ([M − 74]^+^, 4), 178 (2), 164 (5), 149 (8), 123 (18), 109 (40), 95 (65), 81 (93), 74 (25), 67 (99) and 55 (100); ECL 17.93; EI-MS spectrum for 3-pyridylcarbinyl 12,16-heptadecadienoate: 357 ([M]^+^, 7), 356 ([M − 1]^+^, 10), 342 ([M − 15]^+^, 1), 328 ([M − 29]^+^, 2), 316 ([M − 41]^+^, 59), 302 (3), 276 (12), 262 (7), 248 (2), 234 (3), 220 (5), 206 (5), 192 (2), 178 (2), 164 (22), 151 (11), 108 (41) and 92 (100), (Figure 3A and Figure 5A).

Synthetic **1**: EI-MS spectrum for methyl 12,16-heptadecadienoate: 280 ([M]^+^, 4), 248 ([M − 32]^+^, 7), 220 (1), 206 ([M − 74]^+^, 4), 178 (2), 164 (5), 149 (8), 123 (18), 109 (43), 95 (64), 81 (93), 74 (25), 67 (99) and 55 (100); ECL 17.93; EI-MS spectrum for 3-pyridylcarbinyl 12,16-heptadecadienoate: 357 ([M]^+^, 9), 356 ([M − 1]^+^, 12), 342 ([M − 15]^+^, 1), 328 ([M − 29]^+^, 2), 316 ([M − 41]^+^, 69), 302 (2), 276 (12), 262 (8), 248 (2), 234 (3), 220 (5), 206 (4), 192 (2), 178 (3), 164 (20), 151 (10), 108 (34) and 92 (100), (Figure 3B and Figure 5B).

Naturally occurring **2**: EI-MS spectrum for methyl 14,18-nonadecadienoate: 308 ([M]^+^, 4), 276 ([M − 32]^+^, 8), 248 (1), 234 ([M − 74]^+^, 3), 177 (4), 163 (5), 150 (5), 123 (15), 109 (35), 96 (69), 81 (95), 74 (33), 67 (91), 55 (100); ECL 19.93; EI-MS spectrum for 3-pyridylcarbinyl 14,18-nonadecadienoate: 385 ([M]^+^, 9), 384 ([M − 1]^+^, 12), 370 ([M − 15]^+^, 1), 356 ([M − 29]^+^, 2), 344 ([M − 41]^+^, 66), 330 (3), 304 (13), 290 (8), 276 (2), 262 (3), 248 (4), 234 (4), 220 (3), 206 (4), 192 (2), 178 (3), 164 (19), 151 (12), 108 (38) and 92 (100), (Figure 4A and Figure 6A).

Synthetic **2**: EI-MS spectrum for methyl 14,18-nonadecadienoate: 308 ([M]^+^, 4), 276 ([M − 32]^+^, 7), 248 (1), 234 ([M − 74]^+^, 4), 177 (5), 163 (5), 150 (5), 123 (14), 109 (35), 96 (71), 81 (91), 74 (30), 67 (91), 55 (100); ECL 19.93; EI-MS spectrum for 3-pyridylcarbinyl 14,18-nonadecadienoate: 385 ([M]^+^, 10), 384 ([M − 1]^+^, 11), 370 ([M − 15]^+^, 1), 356 ([M − 29]^+^, 1.4), 344 ([M − 41]^+^, 69), 330 (2), 304 (13), 290 (8), 276 (3), 262 (3), 248 (4), 234 (4), 220 (3), 206 (4), 192 (2), 178 (3), 164 (17), 151 (10), 108 (38) and 92 (100), (Figure 4B and Figure 6B).

#### 3.3.1. General Procedure for the Monoprotection of **3a** and **3b**

To a stirred solution of **3a** or **3b** (4.30 mmol) in THF (12 mL) was added NaH (60%, 181 mg, 4.52 mmol) at 0 °C, and the reaction mixture was stirred at 0 °C for 15 min. To the reaction mixture was added MOMCl (0.35 mL, 4.52 mmol), and the resulting mixture was stirred for 16 h at room temperature. The reaction was quenched with saturated NH_4_Cl aqueous solution, and the aqueous mixture was extracted with CH_2_Cl_2_ (3 × 3 mL). The organic extracts were combined, dried, and evaporated to give a colorless oil, which was chromatographed on a SiO_2_ column (15 g, EtOAc/*n*-hexane = 1/10–1/5) to give **4a** [26] or **4b**.

##### 14-(Methoxymethoxy)tetradecan-1-ol (**4b**)

Yield: 53%; mp: 33–35 °C; ^1^H NMR (400 MHz, CDCl_3_) *δ*H: 1.20–1.40 (20H, m), 1.50–1.62 (4H, m), 3.36 (3H, s), 3.51 (2H, t, *J* = 6.6 Hz), 3.64 (2H, t, *J* = 6.6 Hz), 4.62 (2H, s); ^13^C NMR (100 MHz, CDCl_3_) *δ*C: 25.8, 26.3, 29.5, 29.7, 29.8, 32.9, 55.2, 63.1, 68.0, 96.3; IR (KBr): 2920, 2851, 1474, 1464, 1156, 1121, 1109, 1094, 1074, 1055, 1043, 939, 914 cm^–1^; MS (EI): *m*/*z* 274 (M^+^); HREIMS 274.2519 (M^+^), calculated for C_16_H_34_O_3_ 274.2508.

#### 3.3.2. General Procedure for the PCC Oxidation of **4a** and **4b**

To a stirred solution of **4a** or **4b** (2.16 mmol) in CH_2_Cl_2_ (10 mL) were added PCC (837 mg, 3.88 mmol) and NaOAc (424 mg, 5.18 mmol), and the resulting mixture was stirred at room temperature for 16 h. The reaction mixture was filtered through a celite pad and washed with CH_2_Cl_2_ (3 × 3 mL). The filtrate and washings were combined and evaporated to give a black oil, which was chromatographed on a SiO_2_ column (10 g, EtOAc/*n*-hexane = 1/10) to give **5a** or **5b**, which were immediately used for the next Wittig reaction.

#### 3.3.3. General Procedure for the Wittig Reaction of **5a** and **5b**

To a stirred suspension of CH_2_ = CH(CH_2_)_3_P^+^Ph_3_Br^−^ (782 mg, 1.90 mmol) in THF (10 mL) was added a solution of sodium bis(trimethylsilyl)amide (1.9 M in THF, 0.90 mL, 1.71 mmol) at 0 °C, and the resulting suspension was stirred at 0 °C for 5 min. To the suspension was added a solution of **5a** or **5b** (0.95 mmol) in THF (3 mL) at 0 °C, and the resulting suspension was stirred at room temperature for 15 h. The reaction was quenched with saturated NH_4_Cl aqueous solution, and the aqueous mixture was extracted with CH_2_Cl_2_ (3 × 3 mL). The organic extracts were combined, dried, and evaporated to give a colorless oil, which was chromatographed on a SiO_2_ column (15 g, CH_2_Cl_2_/*n*-hexane = 1/5) to give **6a** or **6b**.

##### (*Z*)-17-(Methoxymethoxy)heptadeca-1,5-diene (**6a**)

Yield: 84%; ^1^H NMR (400 MHz, CDCl_3_) *δ*H: 1.20–1.40 (16H, m), 1.50–1.62 (2H, m), 1.95–2.05 (2H, m), 2.05–2.17 (4H, m), 3.35 (3H, s), 3.51 (2H, t, *J* = 6.6 Hz), 4.61 (2H, s), 4.95 (1H, dd, *J* = 11.6, 2.0 Hz), 5.01 (1H, dd, *J* = 17.2, 2.0 Hz), 5.32-5.44 (2H, m), 5.77-5.88 (1H, m); ^13^C NMR (100 MHz, CDCl_3_) *δ*C: 26.2, 26.6, 26.9, 29.3, 29.4, 29.5, 29.6, 29.7, 33.9, 55.0, 67.9, 96.4, 114.5, 128.8, 130.4, 138.5; IR (neat): 2926, 2855, 1461, 1456, 1437, 1150, 1113, 1045, 918, 915 cm^−1^; MS (EI): *m*/*z* 296 (M^+^); HREIMS 296.2720 (M^+^), calculated for C_19_H_36_O_2_ 296.2715.

##### (*Z*)-19-(Methoxymethoxy)nonadeca-1,5-diene (**6b**)

Yield: 94%; ^1^H NMR (400 MHz, CDCl_3_) *δ*H: 1.20–1.40 (20H, m), 1.50–1.62 (2H, m), 1.95–2.05 (2H, m), 2.05–2.17 (4H, m), 3.36 (3H, s), 3.51 (2H, t, *J* = 6.6 Hz), 4.62 (2H, s), 4.96 (1H, dd, *J* = 11.6, 2.0 Hz), 5.02 (1H, dd, *J* = 17.2, 2.0 Hz), 5.32–5.44 (2H, m), 5.77–5.88 (1H, m); ^13^C NMR (100 MHz, CDCl_3_) *δ*C: 26.2, 26.6, 27.0, 29.3, 29.4, 29.5, 29.6, 29.7, 33.9, 55.1, 67.9, 96.4, 114.5, 128.8, 130.5, 138.5; IR (neat): 2926, 2855, 1462, 1458, 1437, 1151, 1113, 1047, 918, 915 cm^−1^; MS (EI): *m*/*z* 324 (M^+^); HREIMS 324.3035 (M^+^), Calculated for C_21_H_40_O_2_ 324.3028.

#### 3.3.4. General Procedure for the Synthesis of **1** and **2**

To a stirred solution of **6a** or **6b** (0.98 mmol) in MeOH (5 mL) was added conc. HCl (5 drops), and the reaction mixture was stirred at 40 °C for 10 h after which it was evaporated to give a pale yellow oil, which was used immediately in the next step. Jones reagent (2.5 M, 0.56 mL, 1.40 mmol) was added to a stirred solution of the oil in acetone (5 mL) at 0 °C, and the resulting mixture was stirred at room temperature for 2 h and then was quenched with MeOH (3 drops), after which the solid was removed through a celite pad and washed with CH_2_Cl_2_ (3 × 3 mL). The filtrate and washings were combined and evaporated to give a pale-yellow oil, which was chromatographed on a SiO_2_ column (20 g, EtOAc/*n*-hexane = 1/20) to give **1** or **2**.

##### (*Z*)-Heptadeca-12,16-Dienoic Acid (**1**)

Yield: 69% in 2 steps; ^1^H NMR (400 MHz, CDCl_3_) *δ*H: 1.20–1.40 (14H, m), 1.63 (2H, quin, *J* = 7.2 Hz), 2.02 (2H, q, *J* = 6.0 Hz), 2.05–2.15 (4H, m), 2.34 (2H, t, *J* = 7.2 Hz), 4.96 (1H, dd, *J* = 10.0, 1.6 Hz), 5.02 (1H, dd, *J* = 17.2, 1.6 Hz), 5.31–5.43 (2H, m), 5.75–5.88 (1H, m); ^13^C NMR (100 MHz, CDCl_3_) *δ*C: 24.7, 26.6, 27.0, 29.0, 29.2, 29.3, 29.4, 29.5, 29.7, 33.7, 33.9, 114.5, 128.8, 130.4, 138.5, 180.2; IR (neat): 2926, 2855, 1713, 1456, 1435, 1418, 1286, 910 cm^−1^; MS (EI): *m*/*z* 266 (M^+^); HREIMS 266.2249 (M^+^), calculated for C_17_H_30_O_2_ 266.2246.

##### (*Z*)-Nonadeca-14,18-Dienoic Acid (**2**)

Yield: 73% in 2 steps; ^1^H NMR (400 MHz, CDCl_3_) *δ*H: 1.20–1.40 (18H, m), 1.63 (2H, quin, *J* = 7.2 Hz), 2.02 (2H, q, *J* = 7.2 Hz), 2.07–2.15 (4H, m), 2.34 (2H, t, *J* = 7.2 Hz), 4.96 (1H, dd, *J* = 9.6, 2.0 Hz), 5.02 (1H, dd, *J* = 19.2, 2.0 Hz), 5.31–5.43 (2H, m), 5.76–5.88 (1H, m); ^13^C NMR (100 MHz, CDCl_3_) *δ*H: 24.7, 26.7, 27.1, 29.0, 29.2, 29.3, 29.4, 29.5, 29.6, 29.7, 33.8, 33.9, 114.5, 128.8, 130.5, 138.5, 180.3; IR (neat): 2918, 2851, 1701, 1470, 1464, 1431, 1412, 1300, 1283, 912 cm^−1^; MS (EI): *m*/*z* 294 (M^+^); HREIMS 294.2563 (M^+^), calculated for C_19_H_34_O_2_ 294.2559.

### 3.4. PPM1A Activation Activity of ***1*** and ***2***

Assays for the activation of PPM1A, formerly PP2Cα, were performed by using recombinant mouse PPM1A and α-casein with the malachite green detection method [27]. Each well of 96-well plates contained 2 μL of FA solution dissolved in MeOH and 48 μL of a typical assay mixture composed of 100 mM Tris-HCl (pH 7.5), 20 mM MgCl_2_, 0.35 mg/mL α-casein, and 16 μg protein/mL of recombinant PPM1A. Reactions were done at 37 °C for 1 h, and then were terminated by adding 100 μL of the malachite green dye solution containing 0.01% (*v*/*v*) Tween 20. After leaving the mixture at room temperature for 10 min, the absorbance value at 650 nm was recorded using a Model 450 Microplate Reader (Infinite F200 PRO, Tecan, Männedorf, Switzerland) [27].

### 3.5. Cytotoxic Activity of ***1*** and ***2*** against HL60 Cells

HL60 cells (RCB0041, RIKEN BioResource Center, Tsukuba, Japan) were grown in RPMI 1640 medium supplemented with 10% heat-inactivated FBS (Biowest SAS, Nuaillé, France) and penicillin (50 units/mL)-streptomycin (50 µg/mL) (Gibco Corp., Carlsbad, CA, USA) in a humidified atmosphere at 37 °C under 5% CO_2_. Cytotoxicity of FAs against HL60 cells was assayed by using the 3-(4,5-dimethylthiazol-2-yl)-2,5-diphenyltetrazolium bromide (MTT) (Dojindo Lab., Kumamoto, Japan) method as described previously [28]. Camptothecin was used as a positive control for HL60 cells with an IC_50_ value of 23.6 nM.

## 4. Conclusions

The structure determination of the previously unreported **1** and **2** in ovaries of the limpet *C. toreuma* was archived by comparison of their ECL values and EI-MS analyses of the synthetic counterparts **1** and **2**. A total synthesis of compounds **1** and **2** was achieved in five-step linear synthetic sequence starting from commercially available diols **3a** and **3b**, respectively. Selective formation of internal *Z*-double bonds was performed by the Wittig reaction under salt-free conditions. Naturally occurring **1** and **2** were finally identified as (12*Z*)-12,16-heptadecadienoic and (14*Z*)-14,18-nonadecadienoic acids, respectively. In addition, both **1** and **2**, as well as structurally related NMI FAs, produced similar PPM1A activation in vitro assay to previously reported 18:1n-9 and 18:2n-6. In the future, to facilitate understanding of their structural diversity and biological properties, additional studies are needed to clarify of the biological functions of **1** and **2,** along with other structurally related NMI FAs, in living organisms.

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
