# Peer review of "Identification and Total Synthesis of Two Previously Unreported Odd-Chain Bis-Methylene-Interrupted Fatty Acids with a Terminal Olefin that Activate Protein Phosphatase, Mg2+/Mn2+-Dependent 1A (PPM1A) in Ovaries of the Limpet Cellana toreuma"

_marinedrugs, 2019, doi:10.3390/md17070410_

Reviewer 1 Report

This paper from Dr Kawashima and coll. contains results of a solid piece of work with an appropriate research design. This research group has investigated for a long time the lipids in ovaries of marine Japanese archeogastropods such as the dominant limpet species Cellana grata or C. toreuma.  The present work seems interesting to get a better knowledge of the physiology and biochemistry of these marine invertebrates.

            Interestingly, the Authors are able to study uncommon very minor non-methylene interrupted (NMI) fatty acids and to show the highly structural diversity of them, including their novel positional isomers. What is exactly the biological role of these novel NMI fatty acids ? And can they have beneficial effects on human health ? From these purposes, the Authors  prepared the adequate quantities of two of them by total organic synthesis in order to investigate their biological properties.

The techniques and experiments used are adequate and nicely illustrated. They are perfectly mastered since they have already been employed in previous works of the Authors.

The organic synthesis of the novel NMI fatty acids, which have a terminal olefin, is well conducted and clearly described. The structural evidences for natural and synthetic compounds are very conclusives.

The objectives of this work are laudable. Within the context the proposed manuscript has undoubtely good originality and technical quality as required for publication in “Marine Drugs“. If the chemical part was written in an excellent manner, I nevertheles would suggest some corrections and improvements regarding the biological activity.

Title

            I would suggest a more precise title, such as:

 “…..Activate PPM1A Protein Phosphatase 1A …“  instead of  “Activate PPM1A“

Introduction and other parts of the manuscript

            1. Indeed, the reader wishes to know for what reasons was this biochemical model chosen?

            2. In the study, the selected molecules activate the chosen enzyme system but what can be the physiological effect and the consequences for human health ? Is this nuclear phosphatase involved here in regulation of cell stress response pathways ? Please, briefly explain the concerned therapeutic area.

            Thus, in order to improve the paper, and to get a better match with the aims and scope of this Journal, I would suggest that the Authors provide some information on these questions.

            In conclusion, I consider this paper accepted  for publication after minor revision.

Author Response

Response to Reviewer 1 Comments

 Comments and Suggestions for Authors (Reviewer 1)

This paper from Dr Kawashima and coll. contains results of a solid piece of work with an appropriate research design. This research group has investigated for a long time the lipids in ovaries of marine Japanese archeogastropods such as the dominant limpet species Cellana grata or C. toreuma.  The present work seems interesting to get a better knowledge of the physiology and biochemistry of these marine invertebrates.

            Interestingly, the Authors are able to study uncommon very minor non-methylene interrupted (NMI) fatty acids and to show the highly structural diversity of them, including their novel positional isomers. What is exactly the biological role of these novel NMI fatty acids? And can they have beneficial effects on human health ? From these purposes, the Authors prepared the adequate quantities of two of them by total organic synthesis in order to investigate their biological properties.

The techniques and experiments used are adequate and nicely illustrated. They are perfectly mastered since they have already been employed in previous works of the Authors.

The organic synthesis of the novel NMI fatty acids, which have a terminal olefin, is well conducted and clearly described. The structural evidences for natural and synthetic compounds are very conclusives.

 The objectives of this work are laudable. Within the context the proposed manuscript has undoubtely good originality and technical quality as required for publication in “Marine Drugs“. If the chemical part was written in an excellent manner, I nevertheles would suggest some corrections and improvements regarding the biological activity.

Title

I would suggest a more precise title, such as:

 “…..Activate PPM1A Protein Phosphatase 1A …“  instead of  “Activate PPM1A“

Response: As suggested, we have revised. (lines 4-5)

Introduction and other parts of the manuscript

1. Indeed, the reader wishes to know for what reasons was this biochemical model chosen?

Response: We are sorry that there was no description about the reason why we focused on PPM1A activation activity. The reason and the references were added in the revised manuscript (lines 75-80, and References 15-21).

 2. In the study, the selected molecules activate the chosen enzyme system but what can be the physiological effect and the consequences for human health ? Is this nuclear phosphatase involved here in regulation of cell stress response pathways ? Please, briefly explain the concerned therapeutic area.

Response: As mentioned above, we added new sentences at line 75. PPM1A is involved in cancer and liver fibrosis, etc..

            Thus, in order to improve the paper, and to get a better match with the aims and scope of this Journal, I would suggest that the Authors provide some information on these questions.

            In conclusion, I consider this paper accepted for publication after minor revision.

Submission Date

11 June 2019

Date of this review

20 Jun 2019 20:50:07

Reviewer 2 Report

In this paper Kawashima et al. describe the isolation of two new very minor odd-chain bis-methylene-interrupted fatty acids, belonging to the class of NMI FAs, from the limpet Cellana toreuma, an organism recently studied by the same group.

These authors have previously published in this field and this study appears a logical continuation of their efforts to find both new representatives of this class of compounds as well as  add new information about their diverse biological activities.

Interestingly, these substances have also been found to activate the PPH1A protein in comparison with other FAs.

ECL values of the methyl esters of the new compounds and GC-MS applied to the 3-pyridyl derivatives were used to suggest their structure that was eventually secured by short syntheses.

The introduction and literature cited appear satisfactory.

In my opinion, just before the discussion about  ECL results (for example before line 168) it would be better to add a short sentence explaining what is the rational of this technique and the pertinent literature. This would help the reader to better understand the following discussion.

The authors claim to have isolated in this study 65 FAs, but no mention is made to them. It would be suitable to shortly mention these substances.

English is to be improved.

Overall this paper deserves publication on Marine Drugs after minor revisions.

Author Response

Response to Reviewer 2 Comments

Comments and Suggestions for Authors (Reviewer 2)

In this paper Kawashima et al. describe the isolation of two new very minor odd-chain bis-methylene-interrupted fatty acids, belonging to the class of NMI FAs, from the limpet Cellana toreuma, an organism recently studied by the same group.

These authors have previously published in this field and this study appears a logical continuation of their efforts to find both new representatives of this class of compounds as well as add new information about their diverse biological activities.

Interestingly, these substances have also been found to activate the PPH1A protein in comparison with other FAs.

ECL values of the methyl esters of the new compounds and GC-MS applied to the 3-pyridyl derivatives were used to suggest their structure that was eventually secured by short syntheses.

The introduction and literature cited appear satisfactory.

In my opinion, just before the discussion about  ECL results (for example before line 168) it would be better to add a short sentence explaining what is the rational of this technique and the pertinent literature. This would help the reader to better understand the following discussion.

Response: Thank you for useful suggestions. We have added the description of ECL values and related literature in the revised manuscript. (lines 175- 177, Reference 22 )

The authors claim to have isolated in this study 65 FAs, but no mention is made to them. It would be suitable to shortly mention these substances.

Response: As suggested, we have revised. (lines 89-91)

English is to be improved.

Response: We have rechecked the revised manuscript.

Overall this paper deserves publication on Marine Drugs after minor revisions.

Submission Date

11 June 2019

Date of this review

24 Jun 2019 12:31:49

Reviewer 3 Report

This paper describes in detail the procedure of identification and synthesis of two novel isomers of odd-chain NMI fatty acids from a limpet Cellana toreuma, as well as the activation of Mg2+/Mn2+-dependent protein phosphatase 1A by these fatty acids. The authors are known for their high professional skills in isolation and structure determination of fatty acids from a limpet Cellana toreuma, and in assessment of their bioactivity. The methods are appropriate and accurately applied, especially as regards the fatty acid analysis, determination of structural features of fatty acids from the mass spectra of their methyl esters and picolinyl esters, and experiments to characterize their biological properties. Although the previous similar studies have already been carried out by the authors, the present study deals with two novel odd-chain NMI fatty acids. The manuscript is well written, and the study is well designed; this is another contribution to our understanding of the diversity of the MNI fatty acids in marine mollusks.

Despite the general positive aspects of the manuscript, there are several issues that need to be improved.

L. 76–77. An explanation is required why the authors chose this enzyme to assess the bioactivity of fatty acids.

L. 80–83. I would recommend rewriting the phrase, which begins with “In this study” and ends with a reference.

In this study, a total of 65 different FAs, along with diverse NMI di-, tri-, and tetraenoic FAs, which we identified previously, were recorded from ovaries of Cellana toreuma, except for minor FAs with less than 0.1% of the total FAs and large amounts of 20:4n-6 and 20:5n-3 (more than 15% of the total FAs) in polar lipids [9].

L. 94–97. Please, revise the fragment “19:2Δ11,18, 9,16-heptadecadienoic acid” in the phrase “In Figure 2, both 1 and 2, as minor and uncommon NMI FA components, were detected along with previously described odd-chain isomers, three NMI heptadecadienoic and two NMI nonadecadienoic acids [12, 13], although a structural analogue of 19:2Δ11,18, 9,16-heptadecadienoic acid (17:2Δ9,16) has been recognized in biological samples for the first time in this study.”

L. 112-113 and 117-188. It is obvious that Figures 3 and 4 are redundant, because the position of the double bonds in the fatty acids is determined from the EI-MS spectra of the picolinyl esters (Figures 5, 6). The information on GS-MS of methyl esters in Section 3.3 is exhaustive.

L. 115, 120, 137, 158. Index “B” should be moved to the end of the phrase: “EI-MS spectra of the methyl esters of naturally occurring 1 (A) and 1 obtained by synthesis (B).”

L. 225. Figure 7. The scale of fatty acid concentrations should be more fractional, at least to plot labels corresponding to the used concentrations on the axis. Also, please, use uniform names for fatty acids (substitute oleic acid on 18:1Δ9 etc.) and name Compound 1 and 2 because their structure are known. The data presented in Figure 7 do not allow estimation of the significance of differences in the activity of different fatty acids. Therefore, it is necessary to indicate the values and SD of fatty acid activities in order to see the significant differences between them.

L. 215–217. “Compounds 1 and 2 showed maximum PPM1A activation activities of 175 ± 7.4% and 169 ± 7.1%, at 100 μM, respectively, compared with oleic and linoleic acids as known PPM1A activators [16].” If so, it looks like the activity of components 1 and 2 is 175% higher than that of oleic.

L. 243–255. It would be better to reduce the description of the commercial sources, as well as the description of the common methods in Section 3.

Author Response

Response to Reviewer 3 Comments

Comments and Suggestions for Authors (Reviewer 3)

This paper describes in detail the procedure of identification and synthesis of two novel isomers of odd-chain NMI fatty acids from a limpet Cellana toreuma, as well as the activation of Mg2+/Mn2+-dependent protein phosphatase 1A by these fatty acids. The authors are known for their high professional skills in isolation and structure determination of fatty acids from a limpet Cellana toreuma, and in assessment of their bioactivity. The methods are appropriate and accurately applied, especially as regards the fatty acid analysis, determination of structural features of fatty acids from the mass spectra of their methyl esters and picolinyl esters, and experiments to characterize their biological properties. Although the previous similar studies have already been carried out by the authors, the present study deals with two novel odd-chain NMI fatty acids. The manuscript is well written, and the study is well designed; this is another contribution to our understanding of the diversity of the MNI fatty acids in marine mollusks.

Despite the general positive aspects of the manuscript, there are several issues that need to be improved.

L. 76–77. An explanation is required why the authors chose this enzyme to assess the bioactivity of fatty acids.

Response: The description of the enzyme was added. (lines 75-80, and References 15-21)

L. 80–83. I would recommend rewriting the phrase, which begins with “In this study” and ends with a reference.

In this study, a total of 65 different FAs, along with diverse NMI di-, tri-, and tetraenoic FAs, which we identified previously, were recorded from ovaries of Cellana toreuma, except for minor FAs with less than 0.1% of the total FAs and large amounts of 20:4n-6 and 20:5n-3 (more than 15% of the total FAs) in polar lipids [9].

Response: As suggested, it was corrected. (line 88)

L. 94–97. Please, revise the fragment “19:2Δ11,18, 9,16-heptadecadienoic acid” in the phrase “In Figure 2, both 1 and 2, as minor and uncommon NMI FA components, were detected along with previously described odd-chain isomers, three NMI heptadecadienoic and two NMI nonadecadienoic acids [12, 13], although a structural analogue of 19:2Δ11,18, 9,16-heptadecadienoic acid (17:2Δ9,16) has been recognized in biological samples for the first time in this study.”

Response: We have revised. (lines 103-104)

L. 112-113 and 117-188. It is obvious that Figures 3 and 4 are redundant, because the position of the double bonds in the fatty acids is determined from the EI-MS spectra of the picolinyl esters (Figures 5, 6). The information on GS-MS of methyl esters in Section 3.3 is exhaustive.

Response: Thank you for your suggestion. As direct visual evidence for diagnostic fragment ions (m/z 81, 95,109, [M-31]+, and M+), Figures 3 and 4 would be useful data for the readers. Also, EI-MS spectra for uncommon NMI FA is very few in the field of marine lipid biochemistry. So these EI-MS spectra are not deleted.

L. 115, 120, 137, 158. Index “B” should be moved to the end of the phrase: “EI-MS spectra of the methyl esters of naturally occurring 1 (A) and 1 obtained by synthesis (B).”

It was corrected, as suggested.

L. 225. Figure 7. The scale of fatty acid concentrations should be more fractional, at least to plot labels corresponding to the used concentrations on the axis. Also, please, use uniform names for fatty acids (substitute oleic acid on 18:1Δ9 etc.) and name Compound 1 and 2 because their structure are known. The data presented in Figure 7 do not allow estimation of the significance of differences in the activity of different fatty acids. Therefore, it is necessary to indicate the values and SD of fatty acid activities in order to see the significant differences between them.

Response: Thank you very much for the valuable suggestions. We revised Fig.7 as you indicated. However, the significance of differences in the enzyme inhibition and/or activation activity are unusual. Thus we would like to remain it. Additionally, we are sorry that Figure 7 was replaced by new one, because SD values were mistaken. As suggested, we changed uniform names for fatty acids, except for compounds 1 and 2, as suggested.

L. 215–217. “Compounds 1 and 2 showed maximum PPM1A activation activities of 175 ± 7.4% and 169 ± 7.1%, at 100 μM, respectively, compared with oleic and linoleic acids as known PPM1A activators [16].” If so, it looks like the activity of components 1 and 2 is 175% higher than that of oleic.

Response: It is correct that the activation activities of compounds 1and 2 at 100 μM are more potent than that of oleic and linoleic acids. However, we think that each value should be judged by the enzyme activation curve at various concentrations.(page 8)

L. 243–255. It would be better to reduce the description of the commercial sources, as well as the description of the common methods in Section 3.

Response: Thank you for useful suggestion. The reduction of the description of the commercial sources and the common methods would be achieved for the answer to the comment of reviewer 3 (lines 247-250).

Submission Date

11 June 2019

Date of this review

25 Jun 2019 07:54:23